# Wave generator as an alternative for classic and innovative wave transmission path vibration mitigation techniques

**Aneta Herbut** [ORCID] *

Wroclaw University of Science and Technology, Faculty of Civil Engineering, Wybrzeże Wyspiańskiego 27, Wrocław, Poland

* aneta.herbut@pwr.edu.pl

## Abstract

In this paper the author proposes an approach in the form of an active wave generator for ground surface vibration reduction. The idea is compared to classic and innovative vibration mitigation techniques. The solution is mainly addressed to prevent people and structures against the destructive effects of anthropogenic vibrations. The efficiency of the presented idea is verified in the paper for two types of excitation–harmonic and impact loads, for points located on the ground surface and below it. The vibration reduction effect for structures is presented in the paper in the case of a three-story building. The advantages and disadvantages of the presented solutions are summarized. Moreover, this paper presents a wide and up-to-date literature review on the vibration control of the ground surface. Classical well-known technologies in the form of ground obstacles are compared with innovative ideas such as metamaterials.

## 1. Introduction

The aim of this paper is to present the author's proposal in vibration mitigation is presented in the form of an active wave generator. The idea is to attenuate vibrations of the ground surface using a new vibration source. It is shown that due to the proper selection of the load characteristics of the wave generator (vibration amplitude and frequency), a vibration reduction can be achieved. Moreover, the paper presents a wide and up-to-date literature review on the vibration control of the ground surface. Classical well-known technologies in the form of ground obstacles are compared with innovative ideas such as metamaterials. While this new idea is currently widely developed, it is often omitted, even in recent studies [1–3]. This is why this issue is emphasized in this paper.

Different methods can be used to prevent the large vibration amplitudes observed in structures. The method selection depends on the situation, specifically the type of structure and the source of the vibration [1–4]. For new structures, it is possible to apply different devices, such as mass dampers, liquid dampers, or frictional dampers [1–6]. However, the application of such devices in existing structures is sometimes difficult, expensive or even impossible. Such cases may appear when this new solution and the additional load it gives were not considered during the design process. On the other hand, instead of modifying the structure

**Data Availability Statement:** All relevant data are within the paper and its Supporting Information files.

**Funding:** The results of numerical calculations presented in this paper have been obtained with the

support of the WCSS Computer Centre, within the frame of Grant No 437.

**Competing interests:** The authors have declared that no competing interests exist.

characteristics (stiffness, damping), the wave energy can be reduced before the wave reaches the structure. Different wave barriers (horizontal, vertical) can be used on the path between the source of vibration and the structure [7–38]. After the wave reaches the ground obstacle, part of the energy is reflected and refracted, and new body waves appear. Ground obstacles–vertical [7–18] and horizontal [19–24]–are well known and commonly used techniques in vibration mitigation. The idea of wave attenuation in soil is currently very popular compared to modifying the structure characteristics. A new stream of research in seismic engineering has appeared recently, as a response of photonic and photonic crystals development. It can be observed that the application of metamaterials in the form of wave barriers and periodic structures/periodic foundations is an important and promising idea for protecting structures and people against the destructive effects of earthquakes or anthropogenic vibrations [27, 28].

All of these classical and new ideas are presented and summarized in this paper, along with their advantages and disadvantages. Moreover, the author's new concept in the form of an active wave generator is proposed and verified for the different load conditions. The proposed solution is addressed to protect structures and people against anthropogenic vibrations. The idea is to generate a new surface wave via an additional vibration source [27, 28]. From the energy requirements, the solution is similar to active dampers attached to structures. However, it does not interfere with the structure, so it is also comparable to wave obstacles or metabarriers in soil. The solution has already been verified for both harmonic and impulse excitations, but only for points located on the ground surface [27, 28]. In the presented paper, the wave generator's efficiency is analysed mainly for points located below the ground surface.

## 2. Wave transmission path vibration mitigation techniques. Literature review

### 2.1. Classical solutions

The simple and most commonly used idea in vibration mitigation is to prepare a vertical barrier between the vibration source and the protected area. Woods first published the results of wide experimental investigations in this area [7]. Analyses of the vibration reduction efficiency of open, empty trenches were performed for the case of a vertical harmonic load applied to the ground surface. The important conclusion yielded from Woods investigations is that a better vibration reduction efficiency can be obtained when the barrier is located closer to the vibration source (active trenches) and has a depth greater than $0.6\lambda_R$. Hence, ground obstacles in the form of vertical barriers can be an effective solution for vibration mitigation, but only in the case of relatively small values of the Rayleigh wavelength (relatively "weak soils" located close to the ground surface and/or large excitation frequencies). By the same geometrical and mechanical features of the barrier, a better vibration reduction efficiency can be obtained in the case of empty trenches (so called "open trenches"), compared to their in-filled counterparts [8–11]. To avoid the instability problems of open trenches, vertical edges of the barrier can be support by concrete elements [12]. A similar beneficial stabilizing effect can be obtained by the use of in-filled trenches. The filling material can have larger elastic moduli values compared to the surrounding soil [13, 14]. A similar reduction effect, observed in the case of infilled vertical barriers, can be obtained by the use of a sheet pile wall or the row of piles [15, 16]. The opposite situation is possible as well, so the obstacle can be filled by a "weaker material", such as Geofoam, water or bentonite (smaller values for the elastic parameters) [8–11, 13, 17, 18]. By placing the material in contact with different characteristics compared to the surrounding medium, the reflection and refraction of the surface waves can be observed. Due to this phenomenon, the scattering effect of the wave energy can be obtained. The interesting idea of an untypical ground barrier in the form of an appropriate arrangement of hills and valleys was

recently presented by Persson et al. [29]. It was proven that the proper arrangement of hills and valleys in the building's vicinity can reduce railway-induced vibration amplitudes by up to 65% of the initial values.

It is not always possible to prepare a vertical obstacle in the soil, or sometimes it is not efficient enough. Horizontal barriers are usually located close to but below the vibration source and are commonly used to protect structures, people and vibration-sensitive equipment in the vicinity of railway tracks or machines. Chouw et al. found that the stiff layer located on the shallow depth can reduce harmonically excited vibrations for vibration frequencies lower than the threshold value [19, 20]. The authors proved that the vibration amplitudes can be significantly decreased by the use of artificial inclusion in the form of a "wave impedance block" (WIB). The most important conclusion yielding from the investigations is that the horizontal stiff ground obstacles can be more effective than vertical trenches with the same dimensions, especially in the case of low excitation frequencies [20, 22]. Takemiya [23] proposed an innovative horizontal barrier similar to the WIB, but it was instead composed of honeycomb elements (HWIB). By the proper arrangement of the soil-cement columns in the honeycomb configuration, significant reduction effect can be observed in the frequency range of 3–5 Hz. It was shown that the reduction efficiency of the honeycomb horizontal barrier is better compared to the conventional countermeasures.

## 2.2. Up-to-date approaches–seismic metamaterials

After semi-active and active approaches, especially in the form of controllable dampers and other devices mounted to the structure, were widely investigated, beginning in the '90s [1–4], a new trend was recently observed. The application of metamaterials to attenuate the structure's vibration is an innovative and promising trend in seismic control. Metamaterials are artificially constructed composite materials composed of a matrix and inclusions. Inclusions are arranged in a periodic way. In the case of phononic crystals, the control of mechanical wave propagation can be obtained by the proper selection of the physical and mechanical properties (density, elastic moduli) between the matrix and inclusions [30–32]. A similar effect is expected in the case seismic waves in soil. Today, the development in seismic metamaterials goes in one of three different directions "seismic invisibility cloak", "Bragg scattering" or "local resonances".

The idea of the "seismic invisibility cloak" is to manipulate the wave path to circumvent the protected area and then to return to the initial path, without a loss in energy. The investigations presented by Diatta and Guenneau [31, 32] are milestones along the way on the possibility of "seismic invisibility cloak" applications to protect structures against earthquakes. The authors proposed spherical and cylindrical cloaks between the vibration source and protected region in an isotropic, homogenous medium [33, 34]. The proposed barrier is able to mitigate P- and S-waves in a low frequency range (0.1 Hz– 10 Hz for the spherical cloak). Colombi et al. [35] presented the idea of a "seismic invisibility cloak" in the form of four cylindrical inclusions around the protected are, placed in a soil matrix. The authors showed that the system of four Luneburg lenses where each lens has a diameter of 150 m is able to protect the structure located inside from the surface wave. The reduction level was equal to 6 dB in the low frequency range of 0–10 Hz. However, the main disadvantage of the solution is that the vibration reduction level is very sensitive to the location of the surface wave source [35]. For the same excitation and the system of lenses rotated by 45 degrees, the effect in the middle of the system was the opposite. The surface wave energy was cumulated in the middle of the system [35]. This is why this idea could not be applied as an "invisibility cloak" for seismic or paraseismic protection, when the source of vibration usually cannot be predicted. The problem of an "invisibility cloak" application for seismic waves is still unsolved and open, unlike the

case to electromagnetic, acoustic or bending waves in thin plates where the explorations are more advanced. From the mathematical point of view, the application of the "invisibility cloak" to seismic waves is much more complicated compared to electromagnetic waves [36, 37]. However, it has to be emphasized that this type of metabarrier has a significant disadvantage, considering applications for the seismic protection of structures. Namely, the barrier is used to manipulate the surface wave propagation, not to dissipate the energy. Even if one selected structure with a special foundation/barrier will be resistant to dynamic excitation, the wave energy is still problematic for other structures behind the one being protected. That is why investigations on the potential applications of metamaterials for seismic protection are currently concerned with a mechanism based on the optimal wave scattering ("Bragg scattering") or the wave energy dissipation by the use of resonators ("local resonances").

Metamaterials for seismic protection can also be designed in such a way to result in the optimal scattering of the surface wave energy ("Bragg scattering") [25, 38–44]. The metabarrier is composed of many empty or in-filled inclusions that are located in the soil matrix and between the vibration source and structure. By the proper selection of elastic moduli and density of those two components (matrix and inclusions), it is possible to create the total barrier for the surface waves. By contact with the inclusions, the surface wave is reflected and refracted, so a reduction effect can be achieved. The scattering effect of the elastic surface wave via the contact with the periodic barrier was described by Meseguer et al. in 1999 [39]. The authors found two regions of reduced vibration in the frequency domain ("band gap"), during an experiment on a marble probe with empty inclusions arranged in a periodic way. A few years later, the experiment on soil-based scattering metamaterial was performed in situ [25]. The publication of Brűlé et al., published in 2014 [25], is usually considered as the breakthrough moment in the method development for seismic protection, presenting the first implementation of a metabarrier in soil. The inclusions were in the form of uniformly distributed holes in the soil matrix. The experimental analyses were supported via numerical validation within the range of the "frequency bandgaps". The reduction level of the proposed solution was even to 20% of the displacements observed before the periodic barrier application, for an excitation frequency of approximately 50 Hz. The investigations performed by Brűlé et al. [25] are a milestone in the seismic protection development, mainly due to the first in situ verification. However, it has to be emphasized that the idea of a seismic metabarrier in the form of inclusions in the soil medium was presented one year earlier [40]. It was proven via numerical analyses that periodically arranged piles in the soil medium can create stopbands for the transmitted waves in the selected range of frequency. The idea of seismic metabarriers based on the optimal scattering of the surface wave ("Bragg scattering") was developed later. Geng Q. et al. proposed a system of two different periodically arranged vertical trenches [41]. This one-dimensional metabarrier was made of vertical rubber and concreate plates. Three wide frequency ranges of attenuated amplitudes were obtained between the excitation frequencies 0–50 Hz. Pu et al. presented a similar idea of a one-dimensional metabarrier that was composed of geofoam trenches located in the soil medium [42]. It was shown that the solution can be effective for the train-induced vibration in the frequency range of 45–60 Hz. Achaoui et al. presented the results of numerical investigations on clamped seismic metabarriers made of concreate or steel with different cross-sections [43]. The main advantage of the clamped barriers is the low frequency range of attenuated amplitudes (the so-called "zero frequency stop bands"). This feature is important because of the low values of the natural frequencies that appear in the case of engineering structures. The effectiveness of metabarriers composed of concreate piles arranged periodically in the layered soil medium was recently investigated by Chen et al. via numerical and experimental approaches [44]. The authors proved that low frequency band gaps can be obtained for both empty (0–4.5 Hz) and in-filled concrete (0–7 Hz) cross-sections.

The different idea is to place special elements in the soil matrix to provoke resonant vibrations of the inclusions ("local resonances") [45–56]. In that way, it is possible to dissipate part of the surface wave energy. Because of the periodic arrangement of inclusions in the soil matrix, this type of barrier is also called a metabarrier. However, in this case the periodic arrangement of inclusions is not necessary to obtain the vibration reduction effect. This is opposite to metabarriers based on optimal wave scattering ("Bragg scattering") or cloaking ("seismic invisibility cloak"), where periodic composition is necessary. A similar idea was already applied to mitigate acoustic waves in an artistic form [45–47]. Muhammad et al. investigated the propagation of surface waves through periodically arranged built-up steel section in the layered soil medium. Due to wave-resonator interaction wide, low frequency bandgaps were observed ($<4$ Hz, $\sim 10$ Hz) [48]. Cacciola et al. presented the results of the numerical and experimental investigations on a single resonant element placed in the soil medium–"vibrating barrier" [49–51]. The reduction level was approximately 40% of the initial vibration amplitudes (without any devices used). However, the dimensions and mass of the element were very large for a single resonant element. A better reduction effect can be obtained by using a whole system of resonating elements. The idea of a metabarrier based on local resonating elements was recently verified by an example of metamaterials existing in nature [26, 51]. The authors showed that a forest, with its periodic arrangement of trees, can be treated as a metamaterial. It was observed that for the selected load frequencies, trees can be excited to resonant vibrations in the longitudinal direction. In that way, the surface wave energy can be dissipated [26, 51]. Two areas of reduced amplitudes were calculated in the frequency domain (approximately 40 Hz and 100 Hz) for the real case of a forest (tree height of 14 m), by the application of the Floquet-Bloch theorem [26]. The results yielding from the numerical investigations were verified by the experiment. It was shown that the reduction efficiency of the proposed idea can be improved by the use of trees with different heights, starting from 1 m to 14 m [51]. The opposite arrangement of resonators (from the highest to the smallest) results in a different mechanism of wave manipulation. By contact with this "metawedge", the surface wave is replaced by a shear wave and its direction on the ground surface opposite. The wide "frequency bandgaps" (30 Hz– 120 Hz) can be obtained by the application of this "metawedge" [51]. Theoretical analyses on the reduction level of the presented solution, based on the Floquet-Bloch theorem, were verified by both numerical [51] and experimental investigations [53, 54]. The main disadvantage of the presented solution is the high frequency range of "bandgaps" compared to the natural frequencies of existing structures. Unfortunately, a natural resonator in the form of a tree has limited dimensions and material properties. However, based on the proposed idea, a series of artificial elements placed in the soil matrix can give the same or even better vibration reduction effects [55, 56]. Palermo et al. presented a metabarrier composed of artificial elements–resonators, located below the ground surface. The artificial resonators were composed of a cylindrical steel mass placed inside the concrete coat. These inclusions were periodically arranged in the soil matrix. The vibration reduction effect of the ground surface was similar to that observed by Colombi et al. [52]. Part of the surface wave energy was transformed to a shear wave and directed deep below the ground surface. The main difference between natural resonators [52] and their artificial counterparts [55, 56] is that these second ones are more flexible in terms of tuning to the low frequencies Moreover, they are placed below the ground surface, which is also an advantage, especially in the case of a built-up area.

## 3. The proposal for vibration mitigation by the use of active wave generator

The aim of this section is to present the author's original approach for vibration mitigation in the form of an active wave generator. The solution is addressed mainly to prevent

anthropogenic ground surface vibrations. By applications of an active wave generator, an additional vibration source is used to generate the new surface wave. The idea is to create a surface wave with a similar vibration frequency and vibration amplitude, but one that has an opposite direction to the wave being attenuated. Summing these two opposite signals allows for a significant reduction in the Rayleigh wave energy (Fig 1). The up-to-date solutions in the form of metabarriers, presented in the previous chapters, are passive methods. Despite the fact that metamaterials can be effective for the entire spectrum of excitation frequencies (not only for the one selected value), the possibility of tuning the material properties to the selected excitation frequency is strongly limited. In some cases, it is better to use active or semi-active systems, which give the possibility of tuning the system to the required excitation frequency [1–4]. The solution in the form of an active wave generator is addressed in such cases, especially

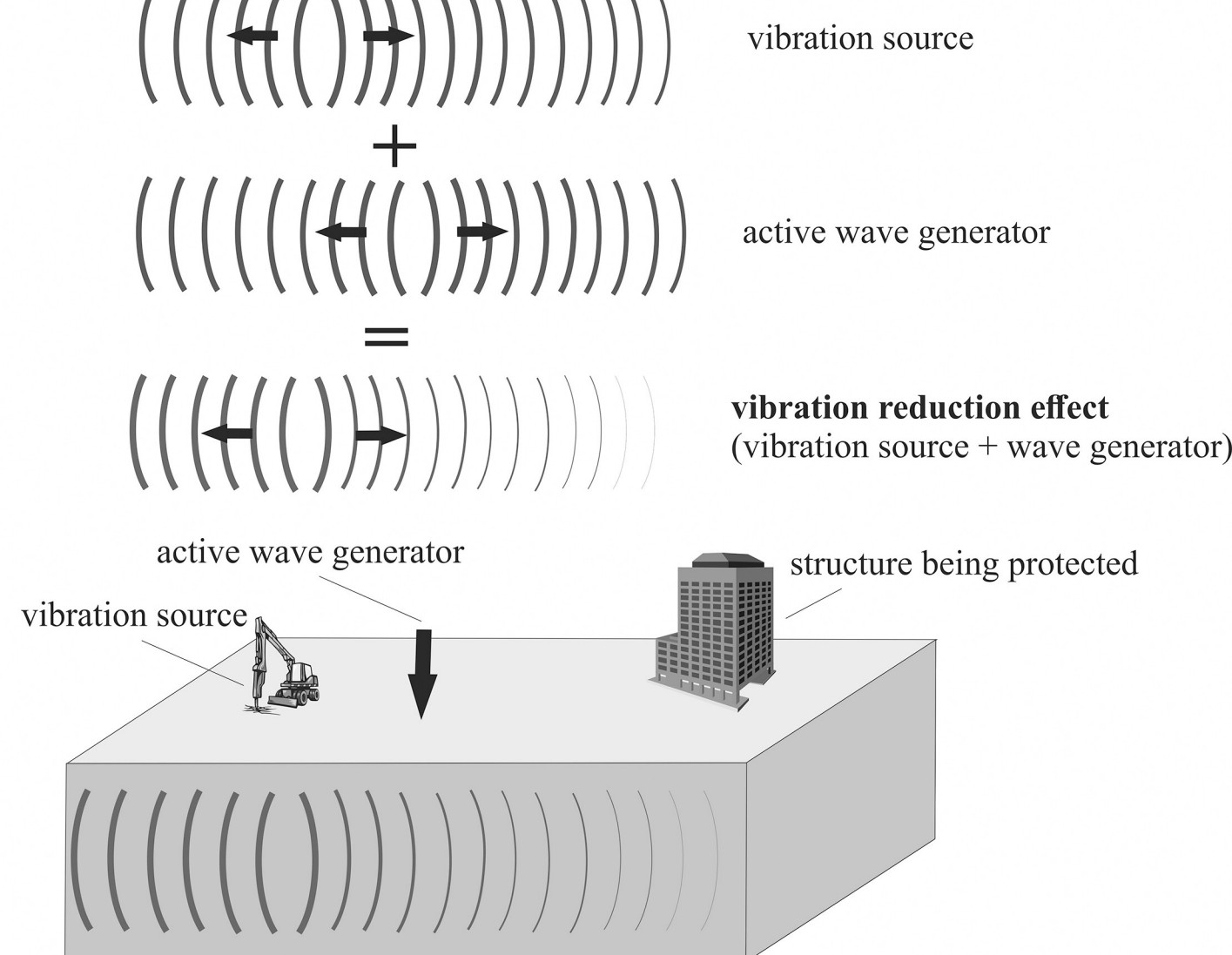

**Fig 1. The idea of the active wave generator.**

in the case of anthropogenic vibrations (machines, geotechnical works). In such situations, it is not reasonable to prepare a barrier or metabarrier in the soil for dynamic protection of the structure. The ideas presented in the previous sections in the form of metabarriers or classical trenches, are addressed for cases when the excitation force acts for a relatively long time or when the time when it appears cannot be predicted. Therefore, the main advantage of the wave generator is that the solution could be cheap, and it could be used immediately and repeatedly in many places and situations. The proposed idea of an active wave generator is similar to solutions already used successfully in ANC (active noise control) systems to attenuate acoustic wave in headphones [57], vehicles [58] or accommodations [59, 60]. It is also comparable to active or semi-active dampers applied to structures [1–4]. In both cases, the energy is added to the system. However, the active wave generator does not interfere with the structure as in the case of active dampers [1–4]. The wave is attenuated before it reaches the structure. That is why, via the use of an active wave generator, not only can one selected building be protected, but the larger area. The efficiency of the active wave generator was already verified in the case of impulse and harmonic excitation, but only for points located on the ground surface and on the structure [27, 28]. In the presented paper, the efficiency of the solution is presented for points located below the ground surface, which is very important in the case of built-up area and structures with underground floors. According to the different European Standards, vibration should be monitored both on the foundation level and on the structure to prevent the vibration's amplifications due to the resonance effect. That is why the additional generators efficiency for a three-story building is also verified. In the presented paper, the efficiency of the wave generator is verified for the two different cases of excitation–a harmonic (Section 3.2.1) and impulse load (Section 3.2.2).

## 3.1. Mathematical model

In this paper, a transversally isotropic soil medium is assumed. The wave propagation phenomenon in a transversally isotropic soil medium with hysteretic damping can be described by the following equations [61]

$$\sum P_x = 0 : \frac{\partial \sigma_x}{\partial x} + \frac{\partial \tau_{yx}}{\partial y} + \frac{\partial \tau_{zx}}{\partial z} + tr \cdot \frac{\partial \sigma_x}{\partial t \partial x} + tr \cdot \frac{\partial \tau_{yx}}{\partial t \partial y} + tr \cdot \frac{\partial \tau_{zx}}{\partial t \partial z} = \rho \frac{\partial^2 u_x}{\partial t^2}$$

$$\sum P_y = 0 : \frac{\partial \tau_{xy}}{\partial x} + \frac{\partial \sigma_y}{\partial y} + \frac{\partial \tau_{zy}}{\partial z} + tr \cdot \frac{\partial \tau_{xy}}{\partial t \partial x} + tr \cdot \frac{\partial \sigma_y}{\partial t \partial y} + tr \cdot \frac{\partial \tau_{zy}}{\partial t \partial z} = \rho \frac{\partial^2 u_y}{\partial t^2}$$

$$\sum P_z = 0 : \frac{\partial \tau_{xz}}{\partial x} + \frac{\partial \tau_{yz}}{\partial y} + \frac{\partial \sigma_z}{\partial z} + tr \cdot \frac{\partial \tau_{xz}}{\partial t \partial x} + tr \cdot \frac{\partial \tau_{yz}}{\partial t \partial y} + tr \cdot \frac{\partial \sigma_z}{\partial t \partial z} = \rho \frac{\partial^2 u_z}{\partial t^2} \quad (1)$$

Where $u_x$, $u_y$ and $u_z$ are the displacement components in the $x$-, $y$- and $z$-directions; $\rho$ is the soil density; $\sigma_x$, $\sigma_y$, $\sigma_z$ and $\tau_{xy} = \tau_{yx}$, $\tau_{xz} = \tau_{zx}$, $\tau_{yz} = \tau_{zy}$ are normal and shear elastic stresses, respectively; $tr$ is the relaxation time $tr = 2\xi/\omega$. The relations for the strains in terms of displacements are assumed as: $\varepsilon_x = \partial u_x/\partial x$, $\varepsilon_y = \partial u_y/\partial y$, $\varepsilon_z = \partial u_z/\partial z$, $\gamma_{xy} = \partial u_y/\partial x + \partial u_x/\partial y$, $\gamma_{xz} = \partial u_z/\partial x + \partial u_x/\partial z$, and $\gamma_{yz} = \partial u_z/\partial y + \partial u_y/\partial z$. For an elastic transversally isotropic material with a horizontal plane of isotropy ($x$-$y$), the elastic strain stress relationship can be presented as

follows [62]

$$\begin{bmatrix} \varepsilon_x \\ \varepsilon_y \\ \varepsilon_z \\ \gamma_{yz} \\ \gamma_{xz} \\ \gamma_{xy} \end{bmatrix} = \begin{bmatrix} C_{11} & C_{12} & C_{13} & 0 & 0 & 0 \\ C_{12} & C_{11} & C_{13} & 0 & 0 & 0 \\ C_{13} & C_{13} & C_{33} & 0 & 0 & 0 \\ 0 & 0 & 0 & C_{55} & 0 & 0 \\ 0 & 0 & 0 & 0 & C_{55} & 0 \\ 0 & 0 & 0 & 0 & 0 & C_{66} \end{bmatrix} \begin{bmatrix} \sigma_x \\ \sigma_y \\ \sigma_z \\ \tau_{yz} \\ \tau_{xz} \\ \tau_{xy} \end{bmatrix}, \tag{2}$$

where $C_{11} = 1/E_x$, $C_{12} = -v_x/E_x$, $C_{13} = -v_z/E_z$, $C_{33} = 1/E_z$, $C_{55} = 1/G_{xz}$, and $C_{66} = 2(1+v_x)/E_x$. $E_x$, $E_y = E_x$ are the Young's moduli in the plane of isotropy ($x$-$y$ plane); $E_z$ is the Young's modulus in the plane perpendicular to the plane of isotropy ($z$ plane). Similarly, $v_x$, $v_y = v_x$ and $v_z$ are the Poisson ratios in the plane of isotropy and in the plane perpendicular to the plane of isotropy, respectively. $G_{xz}$ is the shear modulus in the plane perpendicular to the plane of isotropy.

The absorbing boundary conditions are assumed according to Lysmer and Kuhlemeyer [63]. The normal ($\sigma_x$) and shear ($\tau_{xy}$, $\tau_{xz}$) stress components for virtual dampers "fixed" to the right (along coordinate $x = 70$ m in Fig 2) and the left (along coordinate $x = 0$ in Fig 2) boundaries are given by the formula

$$\sigma_x = a\rho V_x \dot{u}_x, \quad \tau_{xy} = b\rho V_{xy} \dot{u}_y, \quad \tau_{xz} = b\rho V_{xz} \dot{u}_z, \tag{3}$$

where $\dot{u}_x$, $\dot{u}_y$, $\dot{u}_z$ are velocities in the $x$-, $y$- and $z$-directions, respectively, and $a$ and $b$ are parameters introduced to improve the wave absorption at the boundaries in the normal and tangential directions, respectively. Research findings indicate that $a = 1$ and $b = 0.25$ can give a reasonable absorption at the boundary [63, 64]. These values are also assumed in the presented study. $V_x$ denotes the P-wave velocity, and $V_{xy}$ and $V_{xz}$ denote the S-wave velocities. The notation $V_{ij}$ indicates propagation in the $i$-direction and polarization in the $j$-direction. The explanation of the wave propagation phenomenon by different types of anisotropy is omitted here. The problem is widely discussed in the literature [63], including the determination of wave velocities. According to Carcione, the velocities in the plane of isotropy of the transversally isotropic medium are defined as $V_x = V_y = \sqrt{C_{11}/\rho}$, $V_{yz} = \sqrt{C_{55}/\rho}$, $V_{xz} = \sqrt{C_{55}/\rho}$, and $V_{xy} = V_{yx} = \sqrt{C_{66}/\rho}$ [63]. In the plane perpendicular to the plane of isotropy, the velocities can be calculated from the following formulas: $V_z = \sqrt{C_{33}/\rho}$, $V_{zy} = \sqrt{C_{55}/\rho}$, and $V_{zx} = \sqrt{C_{55}/\rho}$ [63].

The normal ($\sigma_y$) and shear ($\tau_{yx}$, $\tau_{yz}$) stress components in the virtual dampers "fixed" to the near and far edges of the analysed domain (along coordinates $y = 0$ and $y = 70$ m, see Fig 2) are expressed as

$$\sigma_y = a\rho V_y \dot{u}_y, \quad \tau_{yx} = b\rho V_{yx} \dot{u}_x, \quad \tau_{yz} = b\rho V_{yz} \dot{u}_z. \tag{4}$$

For the bottom boundary ($z = -35$ m) the damping forces are described by the formulas

$$\sigma_z = a\rho V_z \dot{u}_z, \quad \tau_{zx} = b\rho V_{zx} \dot{u}_x, \quad \tau_{zy} = b\rho V_{zy} \dot{u}_y. \tag{5}$$

Both displacement components are assumed to be zero at the bottom edge of the investigated region.

The transversally isotropic, layered half space is assumed, as presented in Fig 2A. The soil elastic parameters are summarized in Table 1 [65, 66]. The other parameters are as follows: Poisson's ratios: $v_{x,1} = 0.280$ and $v_{z,1} = 0.165$; mass density of the soil $\rho_1 = 2000$ kg m$^{-3}$ and

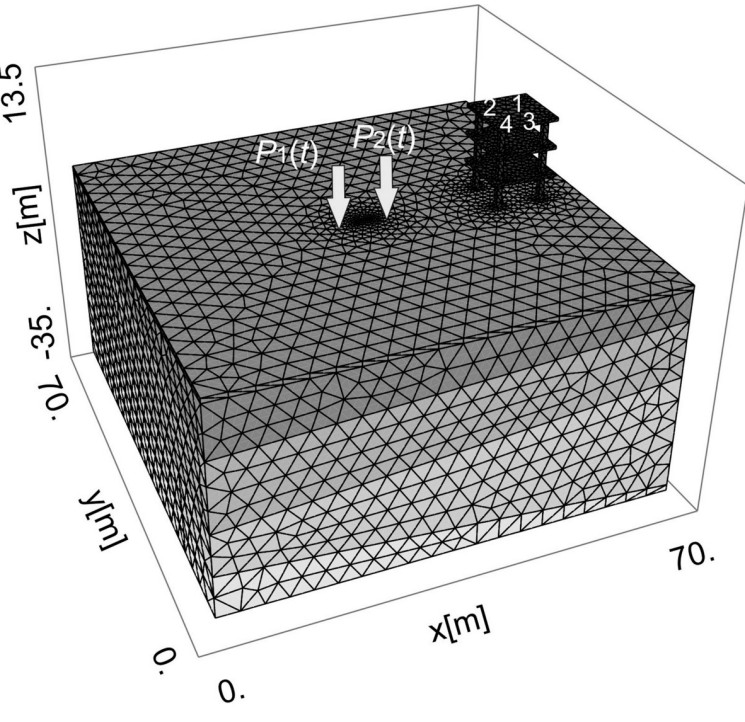

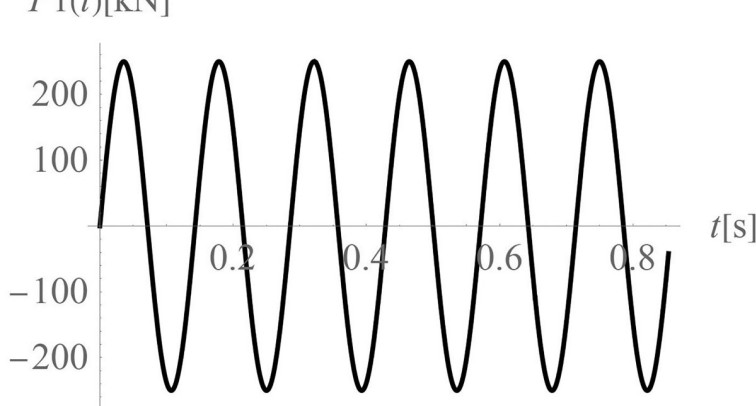

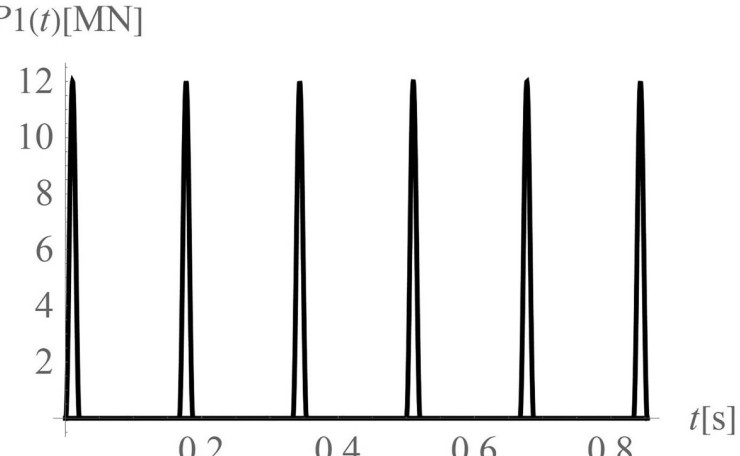

**Fig 2. Considered domain with the vibration source and a wave generator. a. b.** The harmonic excitation force. **c.** The impact excitation force.

damping coefficient $\xi = 1\%$. The boundary surface between the first and second layers is inclined in both directions ($x$ and $y$), and is defined by the following points: $P_1(0,0,-10\text{ m})$, $P_2(70\text{ m},0,-5\text{ m})$ and $P_3(70\text{ m},70\text{ m},-2\text{ m})$. The boundary surface between the second and third layers is inclined, but only in one direction. It is defined by the following points: $P_4(0,0,-21\text{ m})$, $P_5(70\text{ m},0,-15\text{ m})$ and $P_6(70\text{ m},70\text{ m},-15\text{ m})$. The boundary surface between the third and fourth layers is defined by the following points: $P_7(0,0,-28\text{ m})$, $P_8(70\text{ m},0,-34\text{ m})$ and $P_9(70\text{ m},70\text{ m}, -34\text{ m})$. For concrete elements, the isotropic material model is assumed with dynamic parameters: $G_{xz} = 12.2$ GPa, $E_x = E_y = 27$ GPa, $v_x = v_z = 0.167$, $\rho = 2500$ kg m$^{-3}$, and $\xi = 1\%$. The three-story building is located on the right side of the analysed domain at a distance of 20 m from the vibration source $P_1(t)$ (Fig 2). The square plates in each story are 10 m wide and 0.5 m thick, and each story is 4 m high. The concrete square columns (0.8 m wide) are coupled with the concrete square foundation footings, each of which is 3 m wide and 0.8 m thick.

## 3.2. Results of the numerical analyses

To compare the results obtained before and after using the additional generator, the non-dimensional factor *AMF* is introduced after Woods [7]. The amplitude mitigation factor ($AMF_i$, *for i = x,y,z*) in each direction analysed is expressed as

$$AMF_i = \text{Max}(|u_i|)/\text{Max}(|u_{i,o}|). \tag{6}$$

The calculations are performed for two cases: with ($u_i$) and without ($u_{i,o}$) the use of an additional generator. The maximum values of displacements are compared for both cases.

To express the average reduction effect in the selected area, the average amplitude mitigation factor is introduced (7). The average vibration reduction effect in the selected direction $i$ ($i = x,y,z$) for the area located on the right side of the excitation force ($x > 35$ m, $0 < y < 70$ m) can be described as

$$AMF_{av,i} = (1/((x_2 - x_1) \cdot (y_2 - y_1)))\iint_{x_1,y_1}^{x_2,y_2} Max(|u_i|)/Max(|u_{i,o}|)dx\,dy, \tag{7}$$

where $x_1 = 35$ m, $y_1 = 0$, $x_2 = 70$ m, and $y_2 = 70$ m describe the range of the analysed area.

For both situations (with and without the wave generator), the maximum absolute values of the displacement components at each of the 200 points are established, and the AMF is then evaluated. The analyses are made for all three displacement components and are performed for the time interval $4T$. This time is sufficient for the surface waves to cover the distance between the force and the boundary of the analysed region. The appropriate partial differential equations are solved using FlexPDE Professional V6 software (www.pdesolutions.com). Then, the obtained results are analysed using Mathematica 11 software (www.wolfram.com).

In this paper, the efficiency of the wave generator is verified for two different excitation cases of excitation. In the first example, the excitation force is described by the harmonic load (Section 3.2.1), and in the second example an impulse excitation is considered (Section 3.2.2).

**Table 1. Dynamic properties of different soil deposits.**

| Properties | Deposit 1 | Deposit 2 | Deposit 3 | Deposit 4 |
|---|---|---|---|---|
| Dynamic shear modulus $G_{xz}$ [MPa] | $G_{xz,1} = 120.0$ | $G_{xz,1} = 60.0$ | $G_{xz,3} = 280.0$ | $G_{xz,4} = 400.0$ |
| Young's modulus in the plane of isotropy $E_x$ [MPa], $E_y = E_x$ | $E_{x,1} = 463.8$ | $E_{x,2} = 231.9$ | $E_{x,3} = 1082.2$ | $E_{x,4} = 1546.0$ |
| Young's modulus in the plane perpendicular to the plane of isotropy $E_z$ [MPa] | $E_{z,1} = 343.6$ | $E_{z,2} = 171.8$ | $E_{z,3} = 801.6$ | $E_{z,4} = 1145.2$ |

**3.2.1. Case 1—Harmonic excitation.** In this section, the efficiency of the wave generator is verified by harmonic excitation in the case of pile driving. The vibration source in the form of a harmonic load ($P_1(t)$ in Fig 2B) is located in the middle of the considered region ($x = 35$ m, $y = 35$ m, $z = 0$). The excitation force can be described by the following formula

$$P_1(t) = A\sin(2\pi f \cdot t) \cdot (\mathrm{H}(t - t_b) - \mathrm{H}(t - t_e)), \tag{8}$$

where $A = 250$ kN is the amplitude of the excitation, $f = 7$ Hz is the excitation frequency, $\mathrm{H}(t)$ is the Heaviside function, $t_b$ is the time when the exciter begins to work, and $t_e$ is the time when it finishes working. For the analysed example, $t_b = 0$ and $t_e = 4T$, where $T$ is the vibration period. The load is applied to a square concrete element with a width of 0.5 m. An additional generator $P_2(t)$ is used to reduce the vibration amplitudes generated by $P_1(t)$ (Fig 2C). It is located at a distance $r = 2$ m on the right side of the applied load ($x = 35$ m+$r$, $y = 35$ m, $z = 0$). The generated force can be expressed as

$$P_2(t) = -A\sin(2\pi f \cdot t - \varphi) \cdot (\mathrm{H}(t - t_B) - \mathrm{H}(t - t_E)), \tag{9}$$

where the phase shift $\varphi = 2\pi r/\lambda_{R,1}$. $\lambda_{R,1}$ is the Rayleigh wavelength for the first layer, $t_B$ is the time at which the additional generator begins to work and $t_E$ is the time when it finishes working. For the analysed example, $t_B = r/V_{R,1}$ and $t_E = 4T$, where $V_{R,1}$ is the Rayleigh wave velocity for the first layer. The time delay between the two exciters $P_1(t)$ and $P_2(t)$ corresponds to the time required for the Rayleigh wave to cover the distance between the two exciters.

Surface waves are generated due to harmonic load applied to the ground surface. They spread out with a cylindrical wavefront (Fig 3A) [67]. Additionally, the body waves propagate inside the soil medium. A wave generator applied to the ground surface is the new vibration source. The idea is to create a new surface wave with a similar frequency and vibration amplitude, but in the opposite direction to the wave being attenuated. After summing the displacements (or velocities or accelerations) generated by these two vibration sources, a significant vibration reduction effect can be achieved. The vibration attenuation effect can be especially observed on the right side of the analysed region (Fig 3B).

In Fig 4A–4C, the displacement reduction effect for the points located on the ground surface is presented. The results are presented in the form of the dimensionless amplitude mitigation factor (6) for the area where the reduction effect is achieved ($AMF<1$). It can be observed that significant reduction effect is especially achieved on the right side of the analysed region. In the selected areas, the vibration amplitudes after the application of the wave generator are even more than 10 times smaller than the corresponding initial values (Fig 4A–4C). The

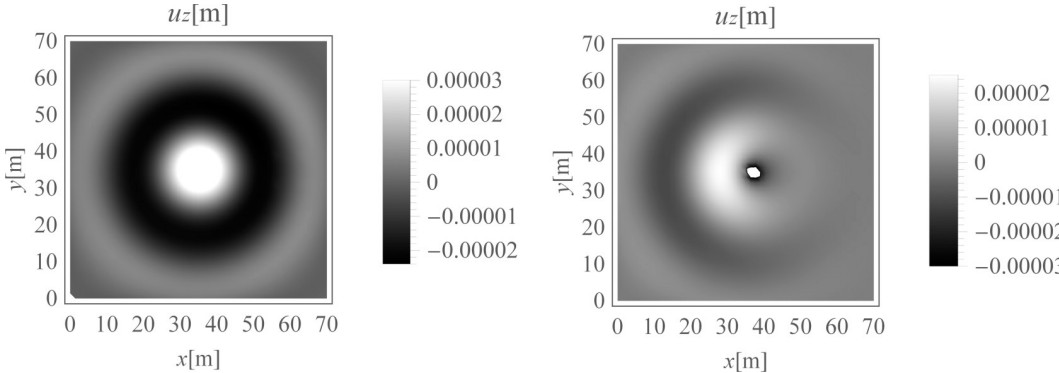

**Fig 3.** Vertical displacement component $u_z$ [m] for the ground surface, for t = 0.21 s; (a) without the wave generator, (b) with the application of the wave generator.

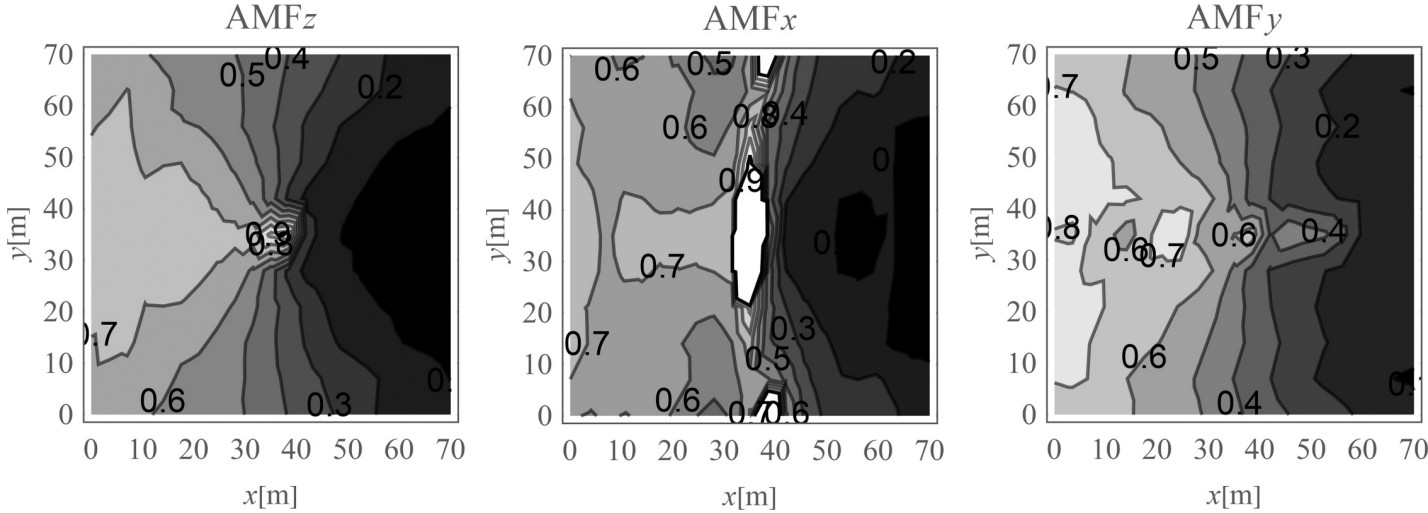

**Fig 4.** Nondimensional amplitude mitigation factor for the ground surface; (a) for the vertical displacement component $u_z$; (b) for the horizontal displacement component $u_x$; and (c) for the horizontal displacement component $u_y$.

average reduction effect for the region located on the right side of the excitation force was also calculated.

The average amplitude mitigation factors $AMF_{av}$ (7) for the points located on the right side of the analysed region ($x > 35$ m, $0 < y < 70$ m) are equal to: $AMF_{av,z} = 0.20$, $AMF_{av,x} = 0.29$ and $AMF_{av,y} = 0.24$ for each displacement component, respectively. These values are addressed to the points located on the ground surface. For underground structures (underground floors, tunnels) it is important to protect the areas located below the ground surface against the destructive effect of a dynamic load. In Figs 5 and 6, the vibration reduction efficiency of the wave generator is presented for the points located below the ground surface. In Fig 5, the values of amplitude mitigation factor are presented for points located 5 m below the ground surface and for the area where the vibration attenuation is achieved (AMF<1). In Fig 6, the AMF

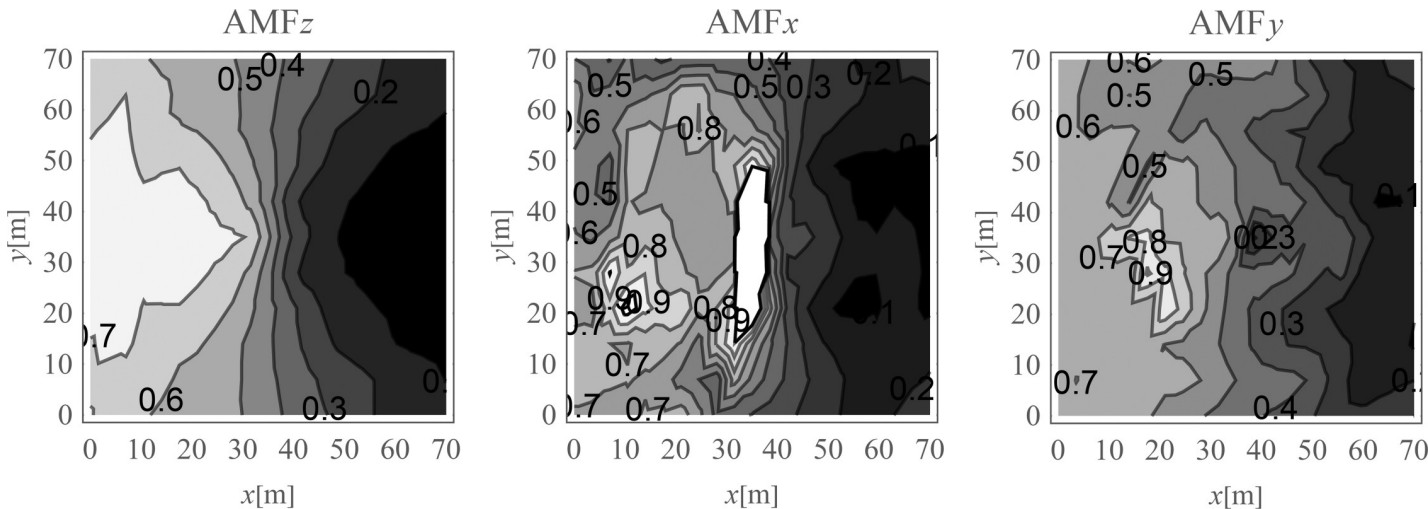

**Fig 5.** Nondimensional amplitude mitigation factor for the points located 5 m below the ground surface; (a) for the vertical displacement component $u_z$; (b) for the horizontal displacement component $u_x$; and (c) for the horizontal displacement component $u_y$.

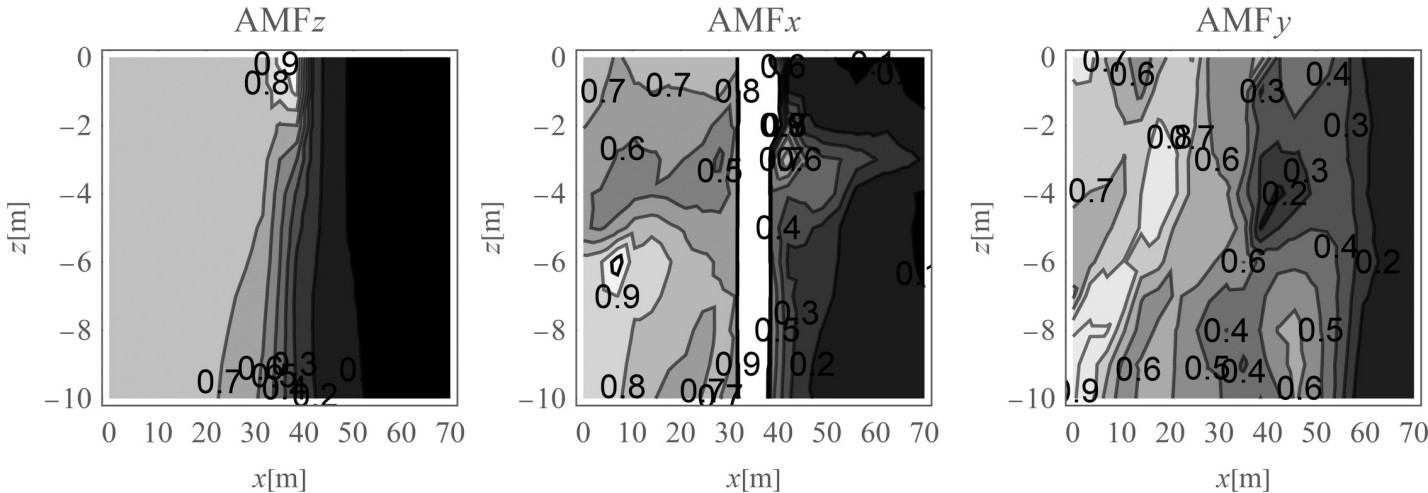

**Fig 6.** Nondimensional amplitude mitigation factor in the vertical cross-section, for the points located in the middle of the analysed area; (a) for the vertical displacement component $u_z$; (b) for the horizontal displacement component $u_x$; and (c) for the horizontal displacement component $u_y$.

values are presented for the vertical cross-section in the middle of the considered region, along the coordinate $y = 35$ m. It can be observed that for each displacement component a significant reduction effect can be achieved. However, the effect is more significant for the vertical displacement component. For the analysed case, this vibration component is associated with the largest values of displacements (Figs 7 and 8). It can be observed that for the horizontal vibration component, the reduction effect is worsened on the right side and improved on the left side of the analysed region near a depth of 4 m (Fig 6B). This is probably caused by the new soil layer that appears in this area (Fig 6, Table 1). Via contact of the surface wave and the body waves with the boundary surface between two layers with different mechanical and physical properties, the wave is reflected and refracted, and new body waves appear [67]. Reaching the ground surface, reflected waves are the source of the new Rayleigh waves. These new waves can make the reduction effect of the wave generator larger or smaller, depending on the soil's mechanical and physical properties. In Figs 7 and 8, the vibration reduction effect observed on the structure is presented, before (grey line) and after (black line) the application of the wave generator. The results are presented for the point located on the foundation level next to the vibration source (Fig 7) and for the point located on structure in the middle of the highest floor (Point 1 in Fig 2), where the largest vibration amplification can be expected (Fig 8). In Table 2, the results for other points located on the highest floor are summarized in the form of the amplitude mitigation factor AMF (6) for three displacement components (Point 1 –Point 4 in Fig 2). All of these points are usually selected to monitor the dynamics effects on the structures [66–70]. It can be observed that for each characteristic point on the structure, significant reduction effect can be achieved.

**3.2.2. Case 2—Impulse excitation.** The aim of this chapter is to verify the wave generator's efficiency in the case of an impulse load, using the example of impact pile driving. A haversine load of amplitude $A = 2$ MN and duration $d = 0.02$ s is applied to the ground surface [71]. The period of the hammer blow is $T = 1$ s. Each separated $i^{th}$ hammer blow is described by the haversine function $P_{1,i}(t)$

$$P_{1,i}(t) = 0.5A(1 - \cos(2\pi(t - t_{b,i})/(t_{e,i} - t_{b,i}))) \cdot (\mathrm{H}(t - t_{b,i}) - \mathrm{H}(t - t_{e,i})),$$

$$P_1(t) = \sum_{i=1}^{n} P_{1,i}(t), \tag{10}$$

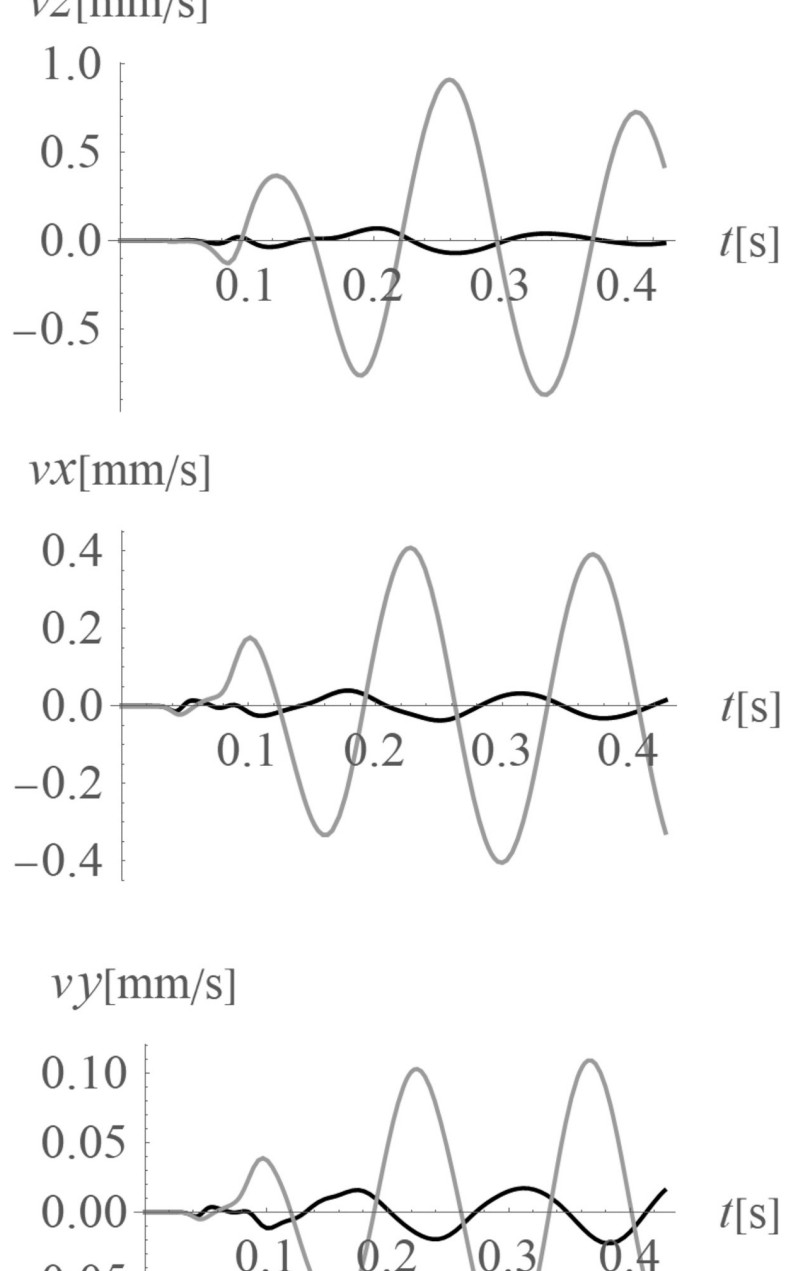

**Fig 7.** Velocities at the foundation level, next to the vibration source; grey line–without the wave generator; black line–after the application of the wave generator; (a) for the vertical velocity component (z-direction); (b) for the horizontal velocity component (x-direction); (c) for the horizontal velocity component (y-direction).

where H($t$) is the Heaviside function, $t_{b,i}$ is the time when the impact begins, and $t_{e,i}$ is the time when it finishes. For the analysed example, $t_{b,i} = (i-1) \cdot T$, $t_{e,i} = t_{b,i}+d$, and $n$ is the number of blows considered. The force is located in the middle of the considered region ($x = 35$ m, $y = 35$ m, $z = 0$) and applied to a square concrete element with a width of 0.5 m. Force $P_1(t)$ is similar

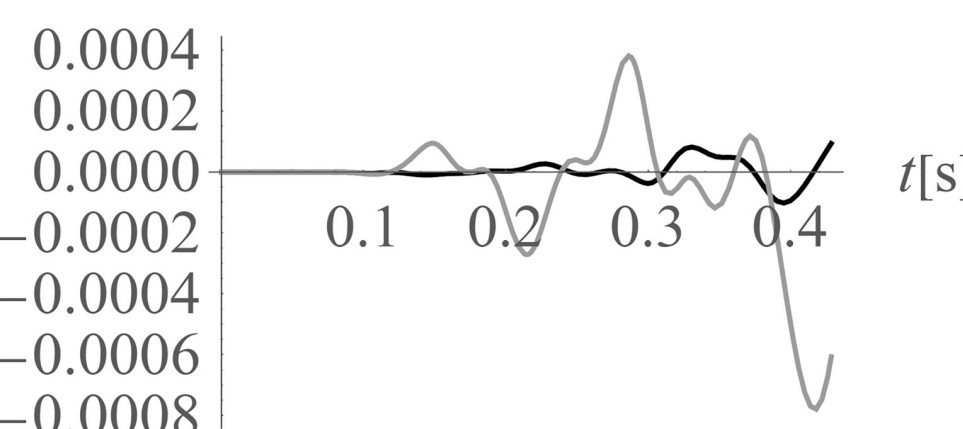

**Fig 8.** Velocities in the middle of the highest floor (Point 1, Fig 2); grey line–without the wave generator; black line–after the application of the wave generator; (a) for the vertical velocity component (z-direction); (b) for the horizontal velocity component (x-direction); (c) for the horizontal velocity component (y-direction).

to $P_2(t)$ but is "moved" along the $t$-axis. The additional generator $P_2(t)$ is used to mitigate the vibration amplitudes generated by $P_1(t)$ (Fig 2), and is located at a distance $r$ on the right side of the applied load ($x = 35$ m$+r$, $y = 35$ m, $z = 0$). The force generated by the wave generator can be expressed as

$$P_{2,i}(t) = -0.5A(1 - \cos(2\pi(t - t_{B,i})/(t_{E,i} - t_{B,i}))) \cdot (\mathrm{H}(t - t_{B,i}) - \mathrm{H}(t - t_{E,i})),$$

$$P_2(t) = \sum_{i=1}^{n} P_{2,i}(t), \tag{11}$$

where $t_{B,i}$ is the time when the impact of the additional generator starts, and $t_{E,i}$ is the time when it finishes. For the analysed example, $t_{B,i} = (i-1)\cdot T + r/V_{R,1}$ and $t_{E,i} = t_{B,i} + d$, where $V_{R,1}$ is the Rayleigh wave velocity for the first layer.

The presented partial differential equations with the appropriate absorbing boundary conditions are solved using FlexPDE Professional V6 software based on the finite element method. Four-node linear tetrahedron finite elements with three degrees of freedom at each node are assumed. A fully bonded soil foundation interface is then assumed.

The idea of the wave generator application in the case of an impulse load is the same as in the case of the harmonic load considered in the previous section (Section 4.2.1). The additional source of vibration acts on the ground surface to cause the new Rayleigh wave. The idea is to create the new surface wave with similar characteristics (frequency, displacements) but directed opposite to the wave being attenuated (Fig 9A). Summing the effects of these two vibration sources allows for a significant vibration reduction effect (Fig 9B).

In Fig 10 the reduction effect of the displacement amplitude in the form of the amplitude mitigation factor (6) is presented for the points located on the ground surface. For each vibration component the attenuation effect is achieved. In the selected areas on the right side of the considered region the vibration amplitudes are reduced by up to 10 times of the initial values.

The average reduction effect (7) for the points located on the ground surface in the area on the right side of the considered region ($x>35$ m, $0<y<70$ m) is $AMF_{av,z} = 0.16$ (for vertical displacement component $z$), $AMF_{av,x} = 0.33$ (for horizontal displacement component $x$), and $AMF_{av,y} = 0.19$ (for horizontal displacement component $y$). The vibration reduction effect for the points located below the ground surface is also examined (Figs 11 and 12) in the presented study. In Fig 11 the amplitude mitigation factor for points located at a depth of 5 m is presented for the region of the reduced vibrations (AMF<1). In Fig 12 the results of similar investigations are presented, but in the middle of the analysed region in the vertical cross-section along coordinate $y = 35$ m. It can be observed that the reduction level of the vertical component is very similar at each considered depth from 0 to 10 m (Figs 10A, 11A and 12A). However, the vibration reduction level of the horizontal displacement components is generally smaller for points located below the ground surface (Figs 11B, 11C, 12B and 12C), compared

**Table 2.** Reduction effect of displacements for the point located on the highest floor (Point 1 –Point 4 in Fig 2).

| AMF | Point 1 | Point 2 | Point 3 | Point 4 |
|---|---|---|---|---|
| $AMF_z$ | 0.066 | 0.069 | 0.058 | 0.078 |
| $AMF_x$ | 0.121 | 0.128 | 0.143 | 0.189 |
| $AMF_y$ | 0.131 | 0.219 | 0.090 | 0.080 |

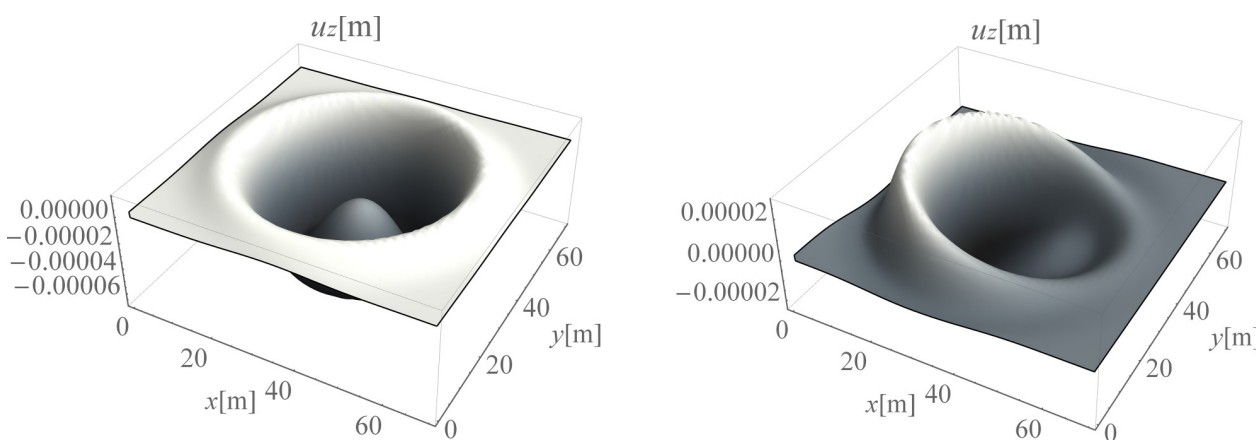

**Fig 9.** Vertical displacement component u$_z$ [m] for the ground surface, for t = 0.15 s; (a) without the wave generator, (b) with the application of the wave generator.

to those located on the ground surface (Fig 10B and 10C). Despite this fact for the analysed cases, a significant mitigation effect can be obtained, especially on the right side from the excitation force (Figs 10 – 12). The reduction effect is more significant for the vertical displacement component. This conclusion is important because the vertical component is the most important in the analysed case because the largest amplitudes of the displacements/velocities/accelerations appear in the z-direction. In Figs 13 and 14 and Table 3 the vibration reduction effect for the structure is summarized after the application of the wave generator. The results are presented for the points located on the foundation level (Fig 13), close to the vibration source, and for four different points located on the highest floor, as in Fig 2 (Fig 14, Table 3). The black line corresponds the case when the wave generator is used, and the grey line corresponds to when it is removed. It can be observed that a significant reduction of the vibration amplitudes is achieved for each analysed point. However, similarly to the results presented for the ground surface, the reduction effect on structure is the most significant for the vertical component. The amplification effect observed in some areas is not a problem in the analysed

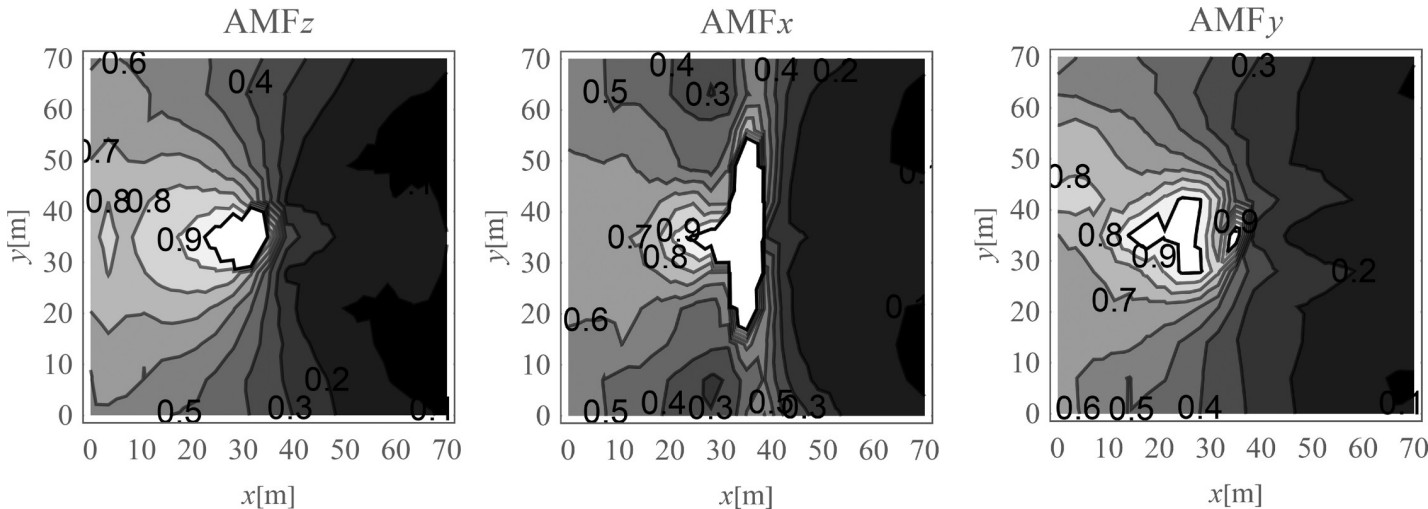

**Fig 10.** Nondimensional amplitude mitigation factor for the ground surface; (a) for the vertical displacement component u$_z$; (b) for the horizontal displacement component u$_x$; and (c) for the horizontal displacement component u$_y$.

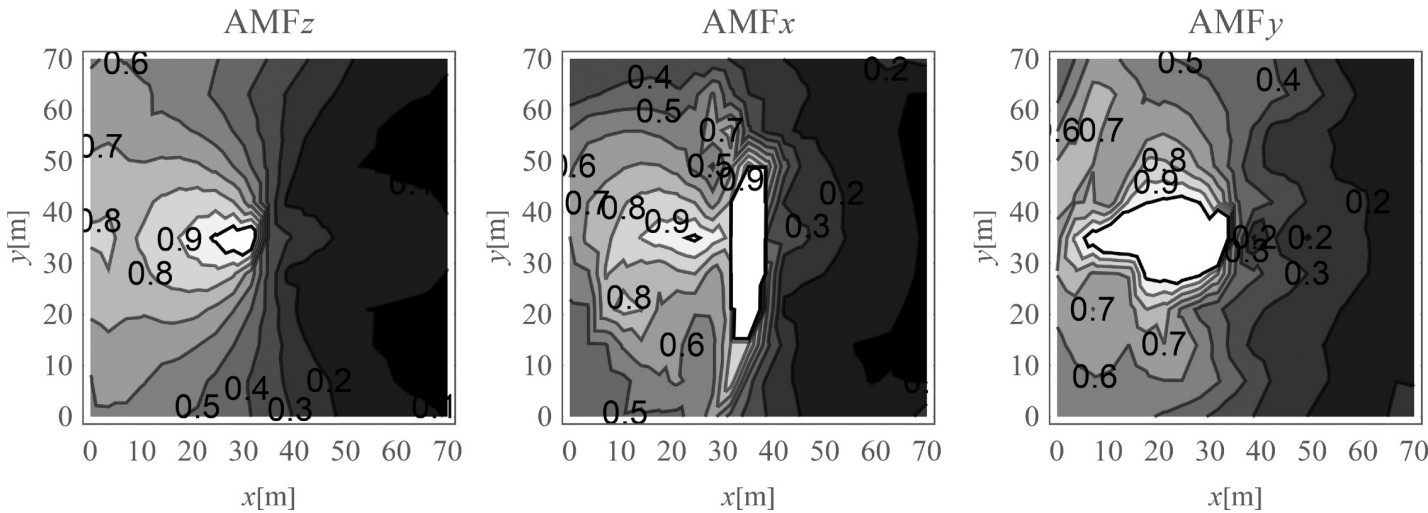

**Fig 11.** Nondimensional amplitude mitigation factor for the points located 5 m below the ground surface; (a) for the vertical displacement component $u_z$; (b) for the horizontal displacement component $u_x$; and (c) for the horizontal displacement component $u_y$.

case. Such an effect can be observed on the left side from the excitation force for the horizontal displacement component $y$ (for 30 m<y<40 m, Figs 11C and 12C). However, the $y$-component of displacement is not significant in the area where the amplification effect is observed. The displacement amplitudes in $y$-direction are very small in this region (Fig 13C) as the horizontal component on the $x$-direction is dominant (Fig 13B).

## 4. Discussion and conclusions

In this paper, new ideas for wave transmission path vibration mitigation techniques were presented, by means of a full-wave numerical demonstration. In the wide literature review classical methods such as trenches were presented, and up-to-date solutions such as metabarriers and periodic systems were emphasized. This part of the paper was important, as a lack of an actual review can be observed, even in the new literature [1–3]. In the second part of the paper,

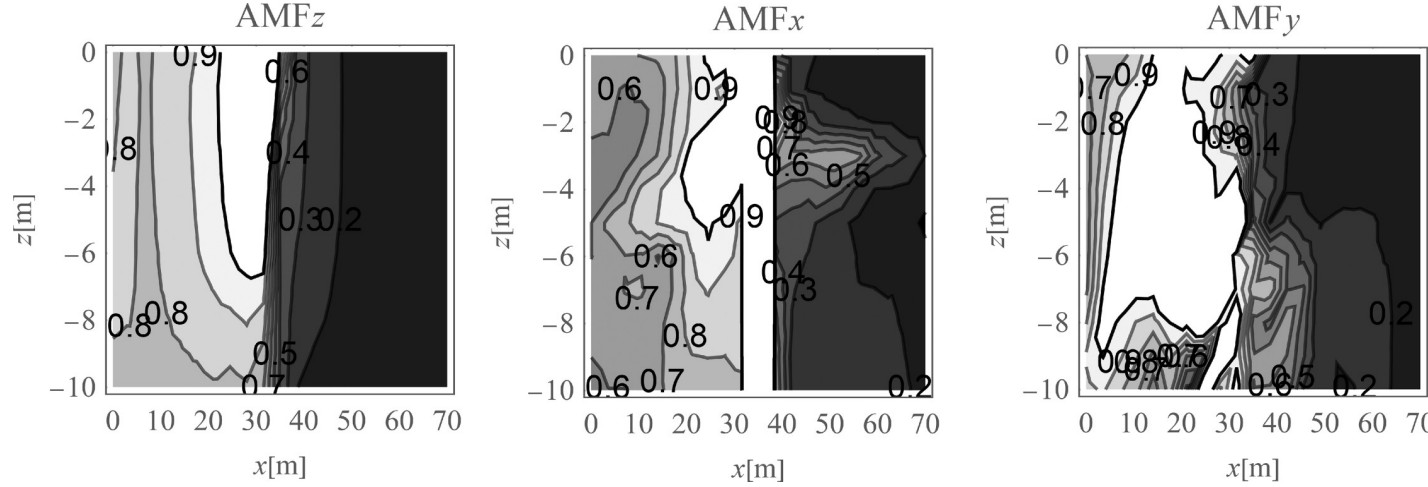

**Fig 12.** Nondimensional amplitude mitigation factor in vertical cross-section, for the points located in the middle of the analyzed (y = 35 m); (a) for the vertical displacement component $u_z$; (b) for the horizontal displacement component $u_x$; and (c) for the horizontal displacement component $u_y$.

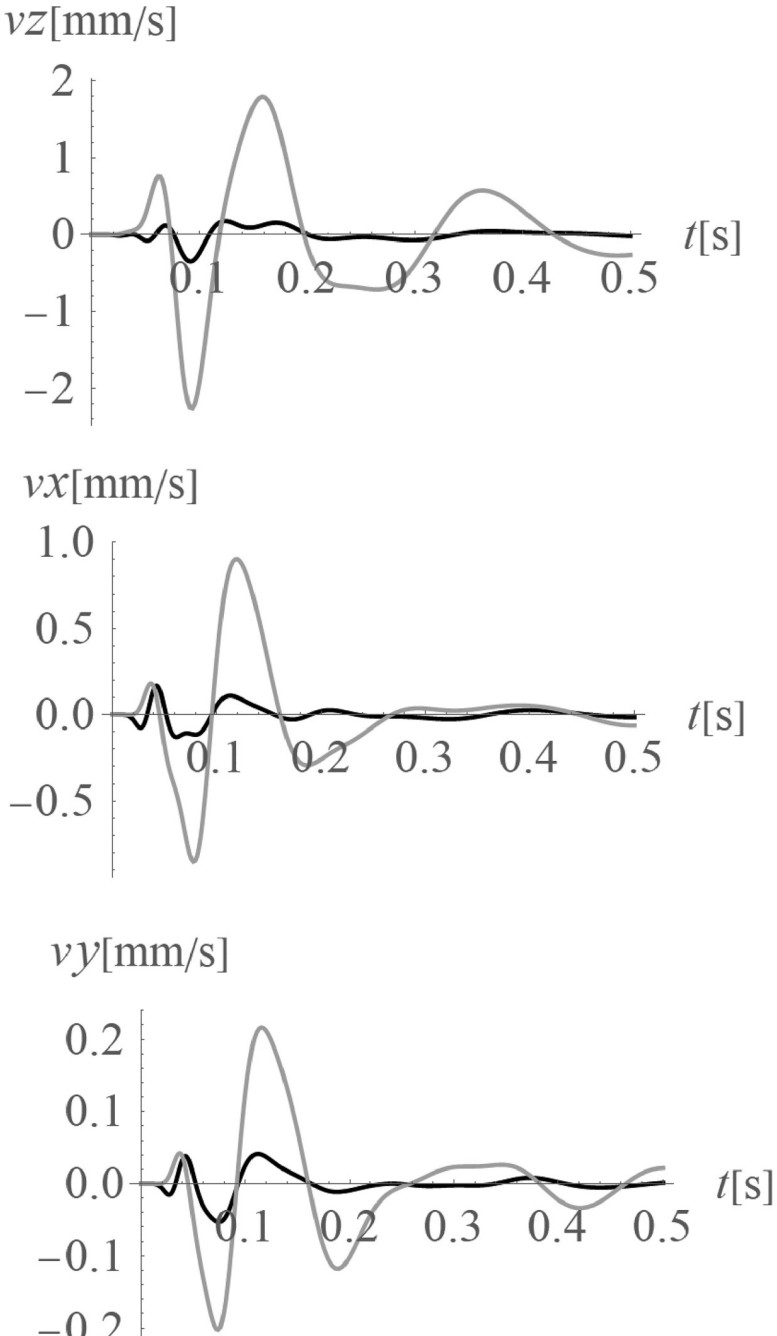

**Fig 13.** Velocities at the foundation level, next to the vibration source; gray line–without the wave generator; black line–after the application of the wave generator; (a) for the vertical velocity component (z-direction); (b) for the horizontal velocity component (x-direction); (c) for the horizontal velocity component (y-direction).

the author's innovative idea in the form of an active wave generator was proposed. The solution was addressed mainly for anthropogenic vibrations. The idea is similar to other active systems for vibration control, such as active dampers mounted to structures or ANC systems, which are currently very popular in acoustics. Compared to active dampers, an active wave generator does not interfere with the structure. This first solution is usually expensive or even

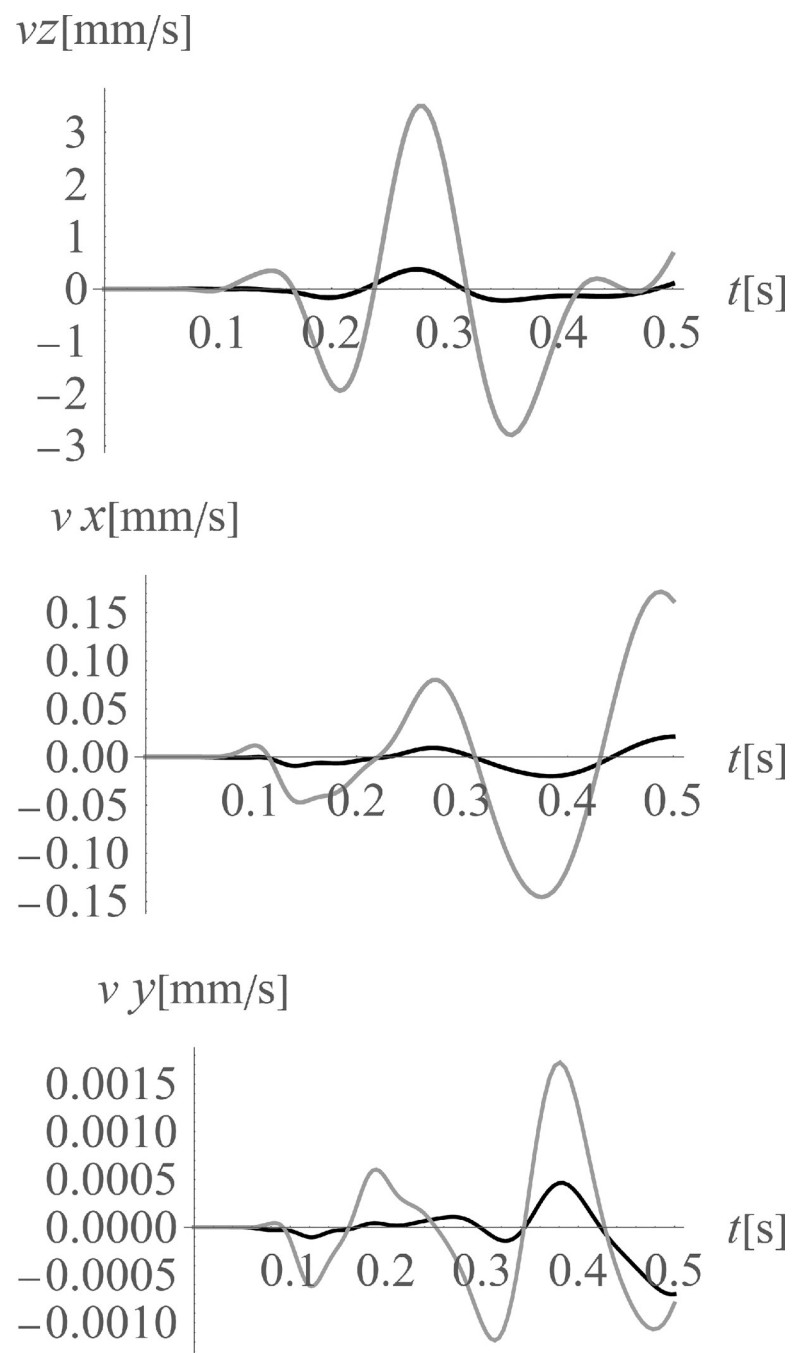

**Fig 14.** Velocities in the middle of the highest floor (Point 1, Fig 2); gray line–without the wave generator; black line–after the application of the wave generator; (a) for the vertical velocity component (z-direction); (b) for the horizontal velocity component (x-direction); (c) for the horizontal velocity component (y-direction).

impossible to implement, especially in the case of existing structures. Moreover, dampers allow for a vibration reduction only for the selected building or even a selected structure element. A wave generator could reduce the vibration amplitudes for a large area. In that way, many structures could be protected. The wave is attenuated here before it reaches the structures, which makes the wave generator similar to soil trenches or metabarriers. However, for

**Table 3. Reduction effect of displacements for the point located on the highest floor (Point 1 –Point 4 in Fig 2).**

| AMF | Point 1 | Point 2 | Point 3 | Point 4 |
|---|---|---|---|---|
| $AMF_z$ | 0.108 | 0.096 | 0.096 | 0.107 |
| $AMF_x$ | 0.124 | 0.125 | 0.130 | 0.129 |
| $AMF_y$ | 0.407 | 0.300 | 0.237 | 0.174 |

some excitation cases or structures, it is not possible or reasonable to build barriers in the soil. Such cases may happen, for example, when there is a load that acts for a relatively short time (like man-made vibrations generated by machines and/or geotechnical works). In such situations, a wave generator can be a cheaper, faster solution. That is the main advantage of the proposed idea. On the other hand, a wave generator adds energy to the system, similar to all active systems used for vibration mitigation. Theoretically, if wrongly used it can make the situation worse. However, with vibration monitoring of structures currently so popular and widely used, such a situation is rather impossible; the system can be simply tuned or switched off when something goes wrong. The efficiency of the proposed solution in the form of an active wave generator was examined in this paper for two cases of excitation–harmonic and impulse loads. The results were presented for the points located on the ground surface and below the ground surface (in the vertical cross-section and at a depth of 5 m). For each analysed case, a significant vibration attenuation effect was achieved. In the selected areas the vibrations were reduced by more than 90% compared to the initial values (AMF<0.1).

In summary, the main advantage of the proposed solution is that it is simple and could be repeatedly used for many different situations. This makes it cheap and fast to implement. The main disadvantage is that, in the selected areas, amplification of the vibration amplitudes is possible. However, it has to be emphasized that a similar intensification effect appears in other methods, such as classical trenches or metabarriers, in the selected areas [7, 25]. A wave generator is an active approach, so it adds energy to the system. That is, why vibration monitoring for the nearby structures is necessary, as it is usually required in the case of anthropogenic vibrations [68–76].

## Supporting information

**S1 Fig. The idea of the seismic invisibility cloak.**
(TIF)

**S2 Fig. The idea of metabarriers based on the Bragg scattering.**
(TIF)

**S3 Fig. The idea of metabarriers based on the local resonances phenomena.**
(TIF)

## Author Contributions

**Conceptualization:** Aneta Herbut.

**Data curation:** Aneta Herbut.

**Formal analysis:** Aneta Herbut.

**Funding acquisition:** Aneta Herbut.

**Investigation:** Aneta Herbut.

**Methodology:** Aneta Herbut.

**Project administration:** Aneta Herbut.

**Resources:** Aneta Herbut.

**Software:** Aneta Herbut.

**Supervision:** Aneta Herbut.

**Validation:** Aneta Herbut.

**Visualization:** Aneta Herbut.

**Writing – original draft:** Aneta Herbut.

**Writing – review & editing:** Aneta Herbut.

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
