## [Decision Letter · Decision Letter 0]

26 Mar 2021

PONE-D-21-04409

WAVE GENERATOR AS AN ALTERNATIVE FOR CLASSIC AND INNOVATIVE WAVE TRANSMISSION PATH VIBRATION MITIGATION TECHNIQUES

PLOS ONE

Dear Dr. Herbut,

Thank you for submitting your manuscript to PLOS ONE. After careful consideration, we feel that it has merit but does not fully meet PLOS ONE’s publication criteria as it currently stands. Therefore, we invite you to submit a revised version of the manuscript that addresses the points raised during the review process.

We look forward to receiving your revised manuscript.

Kind regards,

Yanyu Chen

Academic Editor

PLOS ONE

Journal Requirements:

1. Please ensure that your manuscript meets PLOS ONE's style requirements, including those for file naming. The PLOS ONE style templates can be found athttps://journals.plos.org/plosone/s/file?id=wjVg/PLOSOne_formatting_sample_main_body.pdf andhttps://journals.plos.org/plosone/s/file?id=ba62/PLOSOne_formatting_sample_title_authors_affiliations.pdf

Reviewers' comments:

Reviewer's Responses to Questions

**Comments to the Author**

1. Is the manuscript technically sound, and do the data support the conclusions?

Reviewer #1: Yes

Reviewer #2: Yes

2. Has the statistical analysis been performed appropriately and rigorously? 

Reviewer #1: Yes

Reviewer #2: Yes

3. Have the authors made all data underlying the findings in their manuscript fully available?

Reviewer #1: Yes

Reviewer #2: Yes

4. Is the manuscript presented in an intelligible fashion and written in standard English?

Reviewer #1: Yes

Reviewer #2: Yes

5. Review Comments to the Author

Reviewer #1: Review Comments to the Author

Please use the space provided to explain your answers to the questions above. You may also include additional comments for the author, including concerns about dual publication, research ethics, or publication ethics.:

I have gone through the paper and find that it is suitable for publication as it is with no need for modifications.

Reviewer #2: Dear Author, Congratulation for this paper well written. I suggest few comments but below are general remarks:

The paper will be further improved by a selection of figures relating to the simulations (less figure but more neat – see word file).

6. PLOS authors have the option to publish the peer review history of their article (what does this mean?). If published, this will include your full peer review and any attached files.

Reviewer #1: No

Reviewer #2: **Yes: **Stéphane BRÛLE

---

## [Author Response · Author response to Decision Letter 0]

21 Apr 2021

Thank you for your comments, that turned out to be very helpful in improving my work and which without doubt increased the quality of the paper. The Paper has been revised according to the Reviewer’s comments. All comments with my response are listed below.

1. Section 1. Introduction: the last sentence: “This issue is very important in the case of underground structures (like tunnels) or structures with underground floors.” The sentence is removed in the revised version of the paper as the case of building with underground floors was not considered in the presented paper.

2. Expression “man-made vibrations” is replaced by "anthropogenic vibrations" in the revised version of the paper.

4. Section 3.1 formula (1): ∑𝑃𝑥 = 0 is the equilibrium condition for the infinitesimal soil element. The external load (both forces applied to the ground surface) is defined in the presented mathematical model, as the boundary condition for the selected area located on the ground surface (8) – (11).

5. Damping ratio ksi=1%. I agree that the assumed value is close to the smallest possible damping in soil, however (Das B, Ramana G. Principles of Soil Dynamics. Stamford: Cengage Learning; 2011) gives the values of logarithmic decrement for small strains at about 0.04-0.15 for sand taken from resonant column tests (Das B, Ramana G. Principles of Soil Dynamics. Stamford: Cengage Learning; 2011, Fig. 2.20) and that gives the damping ratio ksi (lower limit) close to 1%. However, during numerical analyses, it can be observed that generally, geometrical damping influences the vibration amplitudes much more than material damping. This second phenomenon does not influence the solution significantly.

6. Table 1: In the presented numerical model a transversally isotropic soil medium is assumed. It gives 5 independent elasticity constants, not 2 as are usually used for isotopic model description (“Wave Fields in Real Media Wave Propagation in Anisotropic, Anelastic ...”J. M. Carcione). That is why there is no simple relationship between E and G in each direction considered. All values of Cij constants are given in the manuscript, C55 can be obtained based on SV wave velocity measurements (SV means wave propagation in x or y direction when particle movement is in z (vertical) direction), since Vsv=sqrt(c55/rho).

7. Soil elastic parameters are taken from literature: Chen X, Birk C, Song C. Numerical modeling of wave propagation in anisotropic soil using a displacement unit-impulse-response-based formulation of the scaled boundary finite element method, Soil Dyn Earthq Eng 2014;65:243–55. https://doi.org/10.1016/j.soildyn.2014.06.019; Das B, Ramana G. Principles of Soil Dynamics. Stamford: Cengage Learning; 2011; Wrana B. Soil dynamics. Computation models. Kraków: Wydawnictwo Politechniki Krakowskiej; 2016.

8. Formulas (6) and (7): the definition of AMF and AMFav has been changed according to the Reviewer’s suggestion.

9. The Reviewer’s comment: “Just for discussion and order of length: for usual buildings, 4 to 12 mm/s could be acceptable. Indeed, for sensitive devices inside the building, lower values may be required.” I agree that from the point of view of the structure’s safety 3mm/s is the lowest PPV limit for vibration-sensitive structures and the low frequency, long-lasting excitation (DIN 4150-3, BS 5228, …). 

However, the acceptable limits can be lower taking into account human comfort, a time when the load acts (day/night) and the building purpose (hospitals, laboratory, etc …). Moreover, much smaller values are acceptable for technical equipment in the laboratory.

10. Section 4: this part of the paper has been improved according to the Reviewer’s suggestions.

11. Reviewer’s suggestion: “To broaden the conclusion, indicate for example that we know how to obtain information on the arrival of a real earthquake (see Mexico City: http://www.iitk.ac.in/nicee/wcee/article/9_vol7_673.pdf). Knowing the incoming seismic signal, could we imagine propagating a counter-signal for this case?”

The solution proposed in the paper is rather addressed to anthropogenic vibrations than to earthquakes. By geotechnical works or machine exploitation, the load is applied directly to the ground surface and the surface wave is dominant (about 67% of the total energy). In such a case it is easy to select the appropriate counter-signal applied to the ground surface that gives the high level of vibration reduction. Moreover, there is no problem with energy supply in the case of anthropogenic vibrations. Theoretically, during earthquakes after the signal detection, it is also possible to select dominant frequencies of the recorded signal and select the counter-signal. Probably, it could work in numerical simulations, however, I am not convinced it could be efficient in real case of earthquake excitation, due to the complex nature of wave propagation in soil caused by earthquake excitation (P-, SH-, SV-waves reflected and refracted from the boundary surface between soil layers). A counter-signal gives the surface wave dominance. Moreover, there are usually problems with energy supply during such an incident.

12. Figure 1: Figure 1 has been improved according to the Reviewer’s suggestions, additional figures Fig. 2b and Fig. 2c present the load applied as the function of time.

13. Section “References” surname has been changed to “Brûlé” in the revised version of the paper.

---

## [Decision Letter · Decision Letter 1]

10 May 2021

WAVE GENERATOR AS AN ALTERNATIVE FOR CLASSIC AND INNOVATIVE WAVE TRANSMISSION PATH VIBRATION MITIGATION TECHNIQUES

PONE-D-21-04409R1

Dear Dr. Herbut,

We’re pleased to inform you that your manuscript has been judged scientifically suitable for publication and will be formally accepted for publication once it meets all outstanding technical requirements.

Kind regards,

Yanyu Chen

Academic Editor

PLOS ONE

**Comments to the Author**

1. If the authors have adequately addressed your comments raised in a previous round of review and you feel that this manuscript is now acceptable for publication, you may indicate that here to bypass the “Comments to the Author” section, enter your conflict of interest statement in the “Confidential to Editor” section, and submit your "Accept" recommendation.

Reviewer #2: All comments have been addressed

2. Is the manuscript technically sound, and do the data support the conclusions?

Reviewer #2: Yes

3. Has the statistical analysis been performed appropriately and rigorously? 

Reviewer #2: Yes

4. Have the authors made all data underlying the findings in their manuscript fully available?

Reviewer #2: Yes

5. Is the manuscript presented in an intelligible fashion and written in standard English?

Reviewer #2: Yes

6. Review Comments to the Author

Reviewer #2: Dear Author, many thanks for having modified your manuscript.

I note you have given a response to all key points.

7. PLOS authors have the option to publish the peer review history of their article (what does this mean?). If published, this will include your full peer review and any attached files.

Reviewer #2: No

---

## [Editor Report · Acceptance letter]

12 May 2021

PONE-D-21-04409R1 

WAVE GENERATOR AS AN ALTERNATIVE FOR CLASSIC AND INNOVATIVE WAVE TRANSMISSION PATH VIBRATION MITIGATION TECHNIQUES 

Dear Dr. Herbut:

I'm pleased to inform you that your manuscript has been deemed suitable for publication in PLOS ONE. Congratulations! Your manuscript is now with our production department. 

Kind regards, 

on behalf of

Dr. Yanyu Chen 

Academic Editor

PLOS ONE